# Biofilm Formation by *Escherichia coli* Isolated from Urinary Tract Infections from Aguascalientes, Mexico

**DOI:** 10.3390/microorganisms11122858

**Published:** 2023-11-25

**Authors:** Flor Yazmín Ramírez Castillo, Alma Lilian Guerrero Barrera, Josée Harel, Francisco Javier Avelar González, Philippe Vogeleer, José Manuel Arreola Guerra, Mario González Gámez

**Affiliations:** 1Laboratorio de Biología Celular y Tisular, Departamento de Morfología, Centro de Ciencias Básicas, Universidad Autónoma de Aguascalientes, Aguascalientes 20100, Mexico; flor.ramirez@edu.uaa.mx; 2Département de Pathologie et de Microbiologie, Centre de Recherche en Infectologie Porcine et Avicole, Faculté de Médecine Vétérinaire, Université de Montréal, Saint-Hyacinthe, QC J2S 7C6, Canada; josee.harel@umontreal.ca; 3Laboratorio de Estudios Ambientales, Departamento de Fisiología y Farmacología, Centro de Ciencias Básicas, Universidad Autónoma de Aguascalientes, Aguascalientes 20100, Mexico; fjavelar@correo.uaa.mx; 4Toulouse Biotechnology Institute, INSA, UPS, Université de Toulouse, 31077 Toulouse, France; vogeleer@insa-toulouse.fr; 5Departamento de Nefrología, Hospital Centenario Miguel Hidalgo, Aguascalientes 20259, Mexico; dr.jmag@gmail.com; 6Departamento de Infectología, Hospital Centenario Miguel Hidalgo, Aguascalientes 20259, Mexico; mariogzg@hotmail.com

**Keywords:** UPEC, biofilm, virulence factors, antimicrobial resistance, multidrug resistance

## Abstract

Uropathogenic *Escherichia coli* (UPEC) strains are among the leading causes of urinary tract infections (UTIs) worldwide. They can colonize the urinary tract and form biofilms that allow bacteria to survive and persist, causing relapses of infections and life-threatening sequelae. Here, we analyzed biofilm production, antimicrobial susceptibility, virulence factors, and phylogenetic groups in 74 *E. coli* isolated from diagnosed patients with UTIs to describe their microbiological features and ascertain their relationship with biofilm capabilities. High levels of ceftazidime resistance are present in hospital-acquired UTIs. Isolates of multidrug resistance strains (*p =* 0.0017) and the *yfcV* gene (*p* = 0.0193) were higher in male patients. All the strains tested were able to form biofilms. Significant differences were found among higher optical densities (ODs) and antibiotic resistance to cefazolin (*p* = 0.0395), ceftazidime (*p* = 0.0302), and cefepime (*p* = 0.0420). Overall, the presence of *fimH* and *papC* coincided with strong biofilm formation by UPEC. Type 1 fimbriae (*p* = 0.0349), curli (*p* = 0.0477), and cellulose (*p* = 0.0253) production was significantly higher among strong biofilm formation. Our results indicated that high antibiotic resistance may be related to male infections as well as strong and moderate biofilm production. The ability of *E. coli* strains to produce biofilm is important for controlling urinary tract infections.

## 1. Introduction

Urinary tract infections (UTIs) are among the most common bacterial infections, and they are a significant public health problem globally. Worldwide, 404.61 million cases, 236,790 deaths, and 520,200 disability-adjusted life years (DALYs) were estimated in 2019 [1], resulting in a high cost of healthcare treatment. Furthermore, UTIs can lead to multiple severe sequelae, including relapse, pyelonephritis with sepsis, renal scarring, and preterm birth [2,3].

Uropathogenic *Escherichia coli* (UPEC), a member of the extra-intestinal pathogenic *E. coli* (ExPEC), is a primary pathogen causing community (80–90%) and hospital-acquired (30–50%) UTIs [4]. These strains harbor a variety of virulence factors in order to establish the infection, including adhesins (i.e., type 1 and P-fimbriae, S/F1C fimbriae, Dr-binding adhesins), toxins (i.e., cytotoxic necrotizing factor and hemolysin), host defense avoidance mechanisms (i.e., group 2 capsule synthesis), and multiple iron acquisition systems (e.g., aerobactin and salmochelin), as well as biofilm formation [5,6,7]. 

In recent years, infections by UPEC have acquired resistance to nearly all antibiotics currently used in clinical practice, leading to ineffective UTI therapy [8]. This increase is related to the excessive use of antibiotics, including broad-spectrum antibiotics such as fluoroquinolones, cephalosporins, and aminoglycosides [9,10,11], high rates of inadequate antibiotic empirical therapies prescribed without antibiotic susceptibility testing [9,12], the overconsumption of indiscriminate antibiotics by the population, the lack of consumer attachment to the medical prescription in the community [13], the regular exposition of humans to antimicrobial resistance through agriculture and foods from animals previously treated by antibiotics, mostly like growth promotors, and the consumption of inadequately treated drinking water [14]. Due to these selection pressures, UPEC could express a multidrug resistance phenotype and higher values of minimal inhibitory concentrations (MICs), enhancing the possibility of treatment failure in UTIs which is an important risk factor for the development of *E. coli* bacteremia, and prolonging morbidity [15,16]. Indeed, the widespread use of fluoroquinolones in community UTI infections, which were previously used as a first-line agent in the therapy of UTIs, is the cause of the continuous increase in resistance to these drugs [9], which are now discouraged as first-line antibiotics in community infections [8,10]. 

UPEC strains can form intracellular bacterial communities (IBCs) within the bladder epithelium, known as biofilms [17,18]. Additionally, UPEC can form quiescent intracellular reservoirs (QIRs) that reside in the underlying urothelium and can trigger reactivation by the exfoliation of superficial epithelial cells, realizing bacteria back into the bladder and acting as a source of recurrent UTIs [7,8,19,20]. These microbial communities are difficult to eradicate because they are poorly responsive to conventional antimicrobial treatment and are more resistant to the host immune response [21,22]. In addition, it has been demonstrated that UPEC biofilms can persist despite treatment with multiple antibiotics [8,20,23,24].

Indeed, the minimal inhibitory concentration of antibiotics against bacteria in biofilms may be 10^1^–10^4^ times higher than that against planktonic cells [22,25]. Furthermore, the proximity of cells within a biofilm can facilitate horizontal gene transfer of genes encoding for antibiotic resistance, enhancing the spread of antimicrobial resistance and virulence properties [26]. Thus, biofilm production is one of the most important virulence factors possessed by UPEC and has become a primary global concern. 

Several virulence factors have been associated with UPEC strong biofilm-producing strains, including type 1 fimbriae [26,27,28], *papC* [29,30,31], *papG* alleles, *sfa/focDE*, *focG*, *sfa* [28,29,32,33], *cnf1* [27,34], *agn43* [2,6,35], *afa*, *fimH* [29,31,36], *traT* [36], hemolysin presence [21,37], *sdiA*, *rcsA,* and *rpoS* genes [38]. Moreover, the relationship between antimicrobial resistance and biofilm-producing on *E. coli* has been studied [21,37,38,39,40]. However, an inconsistent conclusion was found. This study aimed to characterize the biofilm-forming ability of UPEC isolates recovered from human infections and investigate whether there is a link between the ability to form biofilms and the demographic characteristics of the patients and the characteristics of the UPEC strains.

## 2. Materials and Methods

### 2.1. Bacterial Strains and Detection of Phylogroups and Virulence Genes

A total of seventy-four urine cultures were collected from Centenario Hospital Miguel Hidalgo, Aguascalientes, Mexico, in 2013. The hospital is located in Aguascalientes State, a province of Mexico. The cultures were collected anonymously. Only one non-duplicate *E. coli* isolate was used in the present study. The isolation was performed according to the previously described diagnostic criteria for UTIs [41]. The inclusion criteria were patients with UTIs. Their fresh urine samples contained bacterial counts ≥ 10^5^ colony-forming units per milliliter (CFU/mL). Patients with *E. coli* isolated at least 48 h after admission were considered to have a hospital-acquired infection; all other infections were considered community-acquired [42]. Putative *E. coli* was inoculated on MacConkey agar plates and screened for the *uidA* gene by PCR for confirmation, as previously described [43]. 

As previously described, a multiplex PCR phylo-grouping assay was performed to determine the phylogroups [44]. *E. coli* strains H10407 (phylogroup A), E22 (phylogroup B1), CFT073 (phylogroup B2), ECOR 70 (phylogroup C), 042 (phylogroup D), EDL933 (phylogroup E), and ECOR 36 (phylogroup F) were taken as positive controls. Nuclease-free water was used as the negative control. 

*E. coli* isolates were also screened by PCR for the presence of selected virulence genes associated with *E. coli* strains responsible for extra-intestinal infections. Oligonucleotide sequences and PCR conditions are listed in Appendix A. *E. coli* strains J96 (*sfaS*, *hlyA*, and *cnf1*), CFT073 (*uidA*, *papC*, *fyuA*, *chuA*, *kpsMTII*, *yfcV*, and *fimH*), and UTI189 (*vat*) were used as positive controls. Nuclease-free water was used as the negative control. The *fyuA* (yersiniabactin receptor), *sfa* (S fimbriae), *afa*/*dra* (afimbrial adhesion), *kpsMTII* (capsular polysaccharide genes) [45,46], *fimH* (adhesin of type 1 fimbrae), *cnf1* (cytotoxic necrotizing factor), [45], *yfcV* (major subunit of a putative chaperone usher fimbriae) [46], *hlyA* (alfa-hemolysin) [47,48], *agn43* (antigen 43 precursor/major phase-variable outer membrane protein) [49], *papC* (P fimbriae) [45,46,50], and *vat* (autotransporter serine protease toxin) [51,52] genes were investigated. We selected virulence gene markers based on the relationship with the pathogenesis of UTIs and biofilm formation-relatedness. Production of alpha-hemolysin was tested on 5% sheep blood agar. *E. coli* strains were inoculated onto blood agar plates and incubated overnight at 37 °C. Hemolysis was detected by the presence of a lysis zone around the colony [53]. Details and functions of the virulence gene tested are shown in Appendix A. 

### 2.2. Antimicrobial Susceptibility Testing

Susceptibility profiles to different antimicrobial agents were determined by the agar diffusion method [34]. The tested antimicrobial agents were as follows: amikacin (30 µg), gentamicin (10 µg), tobramycin (10 µg), netilmicin (30 µg), ampicillin (10 µg), ampicillin-sulbactam (10/10 µg), amoxicillin-clavulanic acid (20/10 µg), piperacillin-tazobactam (100/10 µg), cefazolin (30 µg), cefotaxime (30 µg), ceftazidime (30 µg), ceftriaxone (30 µg), cefepime (30 µg), trimethoprim-sulfamethoxazole (25 µg), ciprofloxacin (5 µg), levofloxacin (5 µg), norfloxacin (10 µg), nitrofurantoin (300 µg), ertapenem (10 µg), and imipenem (10 µg). *E. coli* ATCC 25922 was used as the negative control. Norfloxacin susceptibility was tested only on thirty-one samples. All other antibiotics were tested in the 74 isolates. Strains were categorized as resistant (R), multidrug-resistant (MDR), or extensively drug-resistant (XDR). Resistant (R) bacteria were defined as those resistant to at least one agent in all classes of antibiotics tested. MDR bacteria were defined as those resistant to at least one agent in three or more antimicrobial classes, and XDR bacteria were defined as non-susceptibility to at least one agent in all but one or two antimicrobial classes tested [54]. 

### 2.3. Biofilm Formation Anlaysis

Biofilm formation was evaluated as described previously [55,56]. Briefly, overnight cultures of *E. coli* in Luria Bertani (LB) broth were diluted 1:100 in M9 minimal medium plus glycerol 0.2% plus minerals (1.16 mM MgSO_4_, 2 µM FeCl_3_, 8 µM CaCl_2_, and 16 µM MnCl_2_) and incubated overnight at 37 °C. These cultures were diluted (1:100) in M9 medium supplemented with glycerol and minerals, and 150µL was aliquoted in triplicate in a sterile 96-well microtiter plate (Costar^®^ 3370, Corning, NY, USA). *E. coli* ATCC 25922 was included in each assay as a positive control, and M9 medium without bacteria was used as a negative control. Following incubation for 24 h at 30 °C, the wells were washed three times carefully with distilled water and dried at 37 °C for 30 min. The wells were stained with 0.1% (*w*/*v*) crystal violet for 2 min, washed three times with distilled water, and dried at 37 °C for 30 min. Cristal violet was resuspended in 150 µL of ethanol/acetone 80:20 (*v*/*v*) solution. Absorbance was measured at 595 nm using a spectrophotometer (Benchmark plus Microplate Reader, BIO-RAD). As previously published, the results were interpreted and grouped into none, weak, moderate, or strong biofilm-producing [57]. Briefly, the optical density of control (ODc) was defined as the mean OD of the negative control. Strains were classified as non-adherent (OD ≤ ODc), weak (ODc < OD ≤ 2 × ODc), moderate (2 × ODc < OD ≤ 4 × ODc), or strong biofilm-producing (ODc > 4 × ODc).

The ability of *E*. *coli* isolates to express a D-mannose-binding phenotype was measured by the ability to agglutinate *Saccharomyces cerevisiae* cells, as described previously [58,59,60]. The agglutination titer was established as the lowest bacterial dilution at which agglutination was observed. If α-D-mannopyranose (5% (*w*/*v*)) (Sigma, St. Louis, MO, USA) inhibited agglutination, yeast agglutination was considered to be due to type 1 fimbriae [58,59,60]. 

The ability to express curli fimbriae was evaluated as previously described [60,61]. Briefly, each strain was spotting on LB agar plates without NaCl containing 0.004% of Congo Red (CR) and 0.002% of Coomassie Brillant Blue G (USB Corporation, Cleveland, OH, USA). Red or pink colonies indicated Congo Red binding after overnight incubation at 30 °C. Cellulose production was determined by spotting the strain on LB agar plates containing 0.02% of calcofluor (Fluorescent Brightener 28; Sigma-Aldrich, St. Louis, MO, USA). The fluorescence of the colonies was measured under UV light illumination at 360 nm (UV transilluminator, INILAB, and TFP-M/WL) after overnight incubation at 30 °C. *E. coli* CFT073 and *E. coli* ATCC 25922 were positive and negative controls, respectively [61].

### 2.4. Statistical Analysis

Statistical analysis was performed using GraphPad Prism 9.4.1 (GraphPad Software, San Diego, CA, USA). A comparison between the patient characteristics (gender and age) and those of the strains (virulence factors, virulence score, phylogroup, and antimicrobial resistance) in hospital- and community-acquired UTIs was performed using the chi-square or Fisher exact test as categorical variables. Mann–Witney U test, two-tailed, was used to compare biofilm formation among hospital- and community-acquired infections as well as non-susceptible vs. susceptible strains as continuous variables. A comparison among the optical densities of strong, moderate, and weak biofilm-producing strains was performed using the non-parametric one-way analysis of variance (ANOVA) with Dunn’s multiple comparison test. *p* values of <0.05 were considered significant. The virulence factor (VF) scores of the strains were reported as the mean ± SD (standard deviation) of the virulence markers the strains possessed. 

## 3. Results

### 3.1. Demographic Characteristics of Patients with UTI Infection and the Relationship between Antimicrobial Susceptibility and Virulence Genes on E. coli Isolates Strains

Table 1 shows the characteristics of *Escherichia coli* isolates from urinary tract infections. A total of 74 urine samples comprised 18 (24.3%) samples from males and 56 (75.7%) samples from females. The prevalence of UTIs was higher in females than in males. Community-acquired infections (68.9%, 51 isolates) were more commonly detected in comparison to hospital-acquired infections (31.1%, 23 isolates) as well, and they were more frequent in children (60.8%, 45 isolates) than in adults (39.2%, 29 isolates). Adult isolates mainly belonged to hospital-acquired infections (56.5%, 16 isolates), in contrast to child isolates, mainly community-acquired infections (68.6%, 35 isolates, *p* = 0.0402, Table 1). The highest prevalence of UTI was found in children (0–12 years group) of female patients (66.1%, 37 isolates), followed by the same age group of males (44.4%, 8 isolates).

All seven phylogroups and cryptic clades were found in the 74 *E. coli* urinary isolates. The phylogroups D (29.7%), B2 (16.2%), and F (14.9%) were the most common, followed by B1 (12.2%), C (12.2%), A (5.4%), E (1.4%), and cryptic clades (8.2%, Table 1). Different phylogroups and cryptic clades were distributed in hospital and community settings, except for phylogroup E, which was only distributed in isolates from hospital-acquired infections. Most of the strains isolated from females belonged to phylogroup D (37.9%), compared to the isolates from males that mainly belonged to phylogroup B2 (27.8%). The majority of the strains isolated from the community (35.3%) and hospital (21.7%) acquired infections belonged to phylogroups D and B1, respectively (Table 1). *E. coli* isolated from children was predominantly distributed in phylogroup D (37.77%) and isolated from adults in phylogroups B1, B2, and D (17.24% in each phylogroup). Interestingly, phylogroup D comprised most of the multidrug-resistant and highly resistant isolates detected (21.3% and 28.4%, respectively).

Furthermore, isolates from hospital-acquired infections showed a higher percentage of resistance to ceftazidime than isolates from community-acquired infections (52.6%/10 isolates vs. 20.5%/9 isolates, *p* = 0.0355, Table 1). Only three strains were considered extensively drug-resistant (XDR) because they were exclusively susceptible to carbapenems. Interestingly, all strains were resistant to nitrofurantoin, unlike non-XDR strains. 

Among antimicrobial susceptibility, more than half of the strains tested in hospital- and community-acquired infections presented multidrug-resistant patterns (60.9%, 14 isolates, and 64.7%, 33 isolates, respectively; Table 1). Sixty-seven isolates (90.5%) were resistant to at least one antimicrobial agent, and forty-seven isolates (63.5%) were multidrug-resistant. Among the 20 different antibiotics evaluated, all 74 *E. coli* isolates were susceptible to the carbapenems tested (ertapenem and imipenem). Many strains were resistant to ampicillin (84.7%) and trimethoprim-sulfamethoxazole (75.7%). High frequencies of resistance were observed to ampicillin-sulbactam (59.5%), amoxicillin-clavulanic acid (57.6%), levofloxacin (54.2%), cefazolin (49.2%), ciprofloxacin (44.6%), and tobramycin (40.9%). Some isolates were resistant to ceftazidime (30.2%), cefotaxime (19.2%), nitrofurantoin (13.5%), netilmicin (9.7%), and amikacin (8.2%) (Figure 1). 

Interestingly, male isolates presented higher resistance to all antimicrobial classes except for carbapenems (Figure 2), with an increased percentage rate higher to 60% on antibiotics such as ceftriaxone (18.8% vs. 66.7%, females vs. males, respectively), ceftriaxone (21.3% vs. 70.6%), cefepime (21.4% vs. 70.6%), tobramycin (22.2% vs. 64.3%), ciprofloxacin (33.9% vs. 77.8%), and levofloxacin (42.9% vs. 82.4%). Overall, multidrug resistance rates (including XDR strains) to antibiotics tested were higher in males than in females (94.4% vs. 53.57%, *p* = 0.0017, Table 2).

Regarding the virulence profile (Table 1), the most common virulence factors found were *fimH* (87.8%), *fyuA* (79.9%), *agn43* (71.6%), *chuA* (66.2%), *papC* (54.1%), and *kpsMTII* (51.4%). Other virulence genes, including *vat* (28.4%), *yfcV* (28.4%), *sfa* (13.5%), *afa/dra* (12.2%), *hlyA* (12.2%), and *cnf1* (4.1%), were also found. When isolates from hospital- and community-acquired infections were compared, the virulence gene *agn43* was prevalent in isolates from community-acquired infections (65.2%, 15 isolates vs. 74.51%, 38 isolates, *p* = 0.022). In contrast, *kpsMTII* was prevalently detected in isolates from hospital-acquired infections (69.6%; 16 isolates vs. 43.1%; 22 isolates; *p* = 0.0341; Table 1). Moreover, a high prevalence of strains isolated from male patients harboring *yfcV* was found (Table 2, *p* = 0.0193).

### 3.2. Biofilm-Forming Abilities 

Biofilm formation by *E. coli* isolates was assessed using a crystal violet assay (Figure 2). All the strains tested were able to form biofilms either at strong (17.6%, 13 isolates), moderate (73.0%, 54 isolates), or weak biofilm-producing way (9.4%, 7 isolates, Figure 3a). Most isolates from hospital-acquired infections were classified as moderate biofilm-producing strains (69.6%, 16 isolates) and community-acquired infections (74.5%, 38 isolates). Hospital-acquired infection isolates showed a higher ability to produce biofilms (OD 0.401 ± 0.045, mean ± SD, Figure 3b) than community-acquired infection isolates (OD 0.366 ± 0.044); however, no statistically significant differences were found. No significant differences were found in comparing biofilm production by strains coming from adults versus those isolated from children (OD 0.378 ± 0.018 and 0.377 ± 0.014, respectively, Figure 3c). In addition, the strains isolated from children were mainly strong biofilm-producing. Similarly, strains isolated from males (OD 0.382 ± 0.030) showed slightly higher biofilm production than those isolated from females (OD 0.375 ± 0.016). Male and female isolates produced moderated biofilms (77.8%, 14/18 isolates; 71.4%, 40/56 isolates, respectively, Figure 3d).

Among antibiotic resistance, stronger biofilm-producing bacteria were more resistant to ampicillin (100%), ampicillin-sulbactam (77%), and amoxicillin-clavulanic acid (80%) antibiotics. Meanwhile, moderate and weak biofilm production presented a higher resistance range to ampicillin (84% and 67%, respectively) and trimethoprim-sulfamethoxazole (78% and 71%, respectively). Furthermore, all XDR isolates were moderately biofilm-producing (Figure 4a). Strong and moderate biofilm-producing strains were mainly MDR strains (69.2% and 61.1%, respectively). In general, weak biofilm-forming strains were more susceptible (28.6%) to different antibiotics than strong (7.7%) and moderate (7.4%) biofilm-producing strains (Figure 4b).

In addition, when we compared optical densities (ODs) among non-susceptible vs. susceptible isolates (Figure 5), we found a higher OD mean among non-susceptible isolates to ampicillin (0.381 vs. 0.2891, *p* = 0.0450, Figure 5a), ampicillin-sulbactam (0.4021 vs. 0.3397, *p* = 0.0317, Figure 5b), piperacillin-tazobactam (0.4364 vs. 0.3651, *p* = 0.0117, Figure 4c), cefazolin (0.3862 vs. 0.2813, *p* = 0.0395, Figure 5e), ceftazidime (0.4119 vs. 0.2911, *p* = 0.0302, Figure 5g), and cefepime (0.4054 vs. 0.2813, *p* = 0.0420, Figure 5i). Moreover, there was a significant difference between strong and moderate biofilm production and resistance to ampicillin-sulbactam (*p* = 0.0072), cefazolin (*p* = 0.0163), norfloxacin (*p* = 0.0479), ceftriaxone (*p* = 0.0424), and cefepime (*p* < 0.001). The highest percentage of strains producing a strong biofilm was observed in phylogenetic group D (30.8%) versus the other phylogenetic groups, even when more than half of weak biofilm-producing strains also belonged to phylogroup D (57.1%, Table 3). Phylogroup B1 also showed a higher frequency in strong (*7.7*%) and moderate (14.8%) biofilm-producing bacteria compared with weak-producing bacteria (0%) (Table 3). 

Strong biofilm-producing strains presented a higher prevalence of the genes *fimH* (84.6%, 11 isolates) and *papC* (76.9%, 10 strains; Table 3). Among moderate biofilm-forming strains, *fimH* (87.0%, 47 isolates), *fyuA* (81.5%, 44 isolates), and *agn43* (72.2%, 39 isolates) were the most prevalent virulence factors. All weak biofilm-producing strains possessed the *fimH* gene (100%, 7 isolates) and had a higher frequency of *fyuA* (85.7%, 6 isolates) and *agn43* (71.4%, 5 isolates) genes. The virulence genes *papC* (76.9%), *yfcV* (38.5%), and *hlyA* (15.5%) were higher in moderate biofilm-producing strains than in weak biofilm-producing strains. However, non-significant differences were found (Table 3). Furthermore, when we compare the optical densities among strong, moderate, and weak biofilm-producing strains with virulence genes, positives versus negative, within a two-way ANOVA with multiple comparisons, we find significant differences among the optical density (OD) of the *sfaS* gene (OD mean, 0.5928 vs. 0.5108, strong vs. weak, respectively, *p* = 0.0042, Appendix A). In addition, phylogroup B1 also showed higher levels of OD compared with other groups (OD mean, 0.6428 vs. 0.5135, phylogroup B1 vs. others, respectively, *p* = 0.0006, Appendix A).

Hemolysis and hemagglutination were used to confirm the production of the virulence factors *α*-hemolysin, type 1 fimbriae (MSHA), and P fimbriae (MRHA). It was found that 67 strains (91%) were hemolytic, including alpha-hemolysis (53 strains/71.6%) and beta-hemolysis (14 strains/18.9%), and a significant association was found between the presence of *hlyA* and hemolysin production (Spearman rank, r = 0.2342, *p* = 0.0446). Hemolysis was found more on moderate biofilm-producing strains (72.2%) versus strong (69.2%) and weak biofilm-producing strains (71.4%, Table 4). Type 1 fimbriae were more prevalent in strong biofilm-producing strains than in weak biofilm-producing strains (84.6% vs. 57.1%, *p* = 0.0349), and P fimbriae were more frequently found among weak biofilm-producing strains versus strong biofilm-producer strains (42.9% vs. 15.4%, Table 4). While the *fimH* gene was detected in 87.8% of the investigated strains, only 74.3% exhibited MSHA. In contrast, MRHA activity was mediated by P fimbriae, S, F1C, and Dr fimbriae. The MRHA phenotype was observed in 25.7% of the strains (Table 4). Of the nineteen strains exhibiting MRHA, nine harbored *papC*, four *sfaS*, and only one *afa/Dr* gene.

The production of curli-, cellulose-, and cellulose-like extracellular materials was analyzed. Curliated bacteria bind the amyloid dye Congo red (CR), which indicates curli and cellulose production [60]. Calcofluor white (CF) is a fluorochrome that binds to polysaccharides with β-1,3 and β-1,4 linkages, such as cellulose, chitin, and succinoglycans [60,61]. In the CR binding assay, 48/74 strains (64.9%) were Congo red positive. The ability to bind CF was less frequent than CR because 41/74 strains (55.4%) were calcofluor-positive, indicating that bacteria produced curli and cellulose (Table 4). Cellulose and curli production were observed simultaneously. Curli fimbriae were more frequently found among moderate biofilm-producing strains when we compared them to strong and weak biofilm-producing strains (Table 4), and significant differences were found among moderate vs. weak producing strains (68.5% vs. 42.9%, respectively, *p* = 0.0477). Additionally, cellulose was higher in strong (69.2%) and moderate (55.5%) biofilm-producing strains compared to weak (25.7%) biofilm-producing strains (55.5% vs. 25.7%, *p* = 0.0253, Table 4).

## 4. Discussion

In this study, the incidence of UTIs was higher in females than in males, as previously observed [1,3,62,63,64]. Remarkably, the incidence was higher in females from 0 to 12 years, suggesting that in this geographical area, children are more likely to develop UTIs than in other age groups. Moreover, *E. coli* isolated from children was predominantly distributed in phylogroup D (37.77%). This phylogroup comprises multidrug-resistant and highly resistant isolates. Overall, rates of multidrug-resistant strains were higher in males than females and were mainly acquired through the community (55.5%, 10/18 isolates). These observations were like those of Tabasi et al., 2015 [62] and Jombo et al., 2011 [64].

Furthermore, the prevalence of the *yfcV* gene was higher in male infections. This gene encodes the major subunit of a putative chaperone, usher fimbria, associated with UPEC strains with a better ability to colonize the bladder and are considered highly pathogenic strains [46]. In our study, isolates from male patients also had higher levels of antimicrobial resistance to almost all antibiotics tested than female patients. This is in accordance with the study of Gu et al., 2022 [65], where higher susceptibility rates were shown in female infections caused by *E. coli*. This suggests that male infections are challenging to eradicate and advises that local studies must focus on male gender infections to contain UTIs and the spread of antibiotic resistance in UPEC.

The incidence of antimicrobial resistance among *E. coli* strains that cause UTIs is increasing. In this study, most strains were resistant to ampicillin (84.7%) and trimethoprim-sulfamethoxazole (75.7%). These results are similar to those of previous studies [29,31,63]. The increasing resistance of *E. coli* to trimethoprim-sulfamethoxazole can be explicated by the frequent use of this antimicrobial agent, which is recommended as the first-line antibiotic for empirical therapy of uncomplicated acute cystitis [66]. 

In our study, resistance to cephalosporin antibiotics ranged from 19.2% to 49.2%. Similar results were reported by Yilmaz et al. (2016) [67]. In addition, ceftazidime resistance was significantly higher in hospital-acquired infections. Because of the high antimicrobial resistance recorded in the present study, the choice of drugs for prophylaxis of UTIs, such as cephalosporins, mainly in hospital-acquired infections, should be carefully considered. Among fluoroquinolones such as levofloxacin (54.2%) and ciprofloxacin (44.6%), moderate resistance was found. Thus, this might also be due to the abuse of fluoroquinolones on uncomplicated UTI infections, mainly in the community; however, we did not find differences across the hospital and community, neither in ciprofloxacin (52.17% vs. 41.2%, hospital and community, respectively) or levofloxacin (55% vs. 53%, hospital and community, respectively). Precautions need to be taken against using these drugs due to the widespread resistance to antibiotic classes on UPEC. Nitrofurantoin, netilmicin, and amikacin may be better alternatives for treating urinary tract infections in this area. In the present study, 13.5% of the strains were nitrofurantoin resistant, which is concerning because this finding showed increased nitrofurantoin resistance through UTIs. Furthermore, 4.05% (3/74 isolates) of the strains were categorized as XDR isolates; remarkably, all the strains presented a nitrofurantoin-resistant pattern. This is consistent with Khamari et al. (2021) [68], who reported that resistance to nitrofurantoin indicates the underlying extensively drug-resistant (XDR) phenotype in *Enterobacteriaceae*, which could complicate the treatment of UTIs.

Previous studies have shown that *E. coli* strains causing UTIs predominantly belong to phylogroup B2 and, to a lesser extent, phylogroup D [53], and common commensal strains belong to phylogroup A or B1 [45,69]. In a study by Iranpour et al. (2015) [69] and Mirzahosseini et al. (2023) [70], approximately 25% of the *E. coli* isolates from UTIs belonged to the new phylogroups E, F, and cryptic clades; however, they found that phylogroup F had a prevalence of 2.9% of the cases. Another study developed by Ballén et al. (2021) [29] found a prevalence of 6.9% in the phylogroup F and 1.3% in the unknown phylogroup. This is in contrast to our study, where the F group was one of the most prevalent phylogroups (14.9%). These differences in the prevalence of the phylogenetic groups reported in different studies may be explained by differences in sampling location, health status, dietary and host genetic factors, and host social conditions [68,69]. 

Regarding biofilm formation, all the strains tested were able to form biofilms, which is consistent with Ponnusamy et al., 2012 [18] and Dossouvi et al., 2023 [21], who reported a 100% prevalence of biofilm formation in UPEC. Other studies have reported a prevalence of 78% [71], 80% [31,72], and 84.3% [73]. Most strains were able to form moderate biofilms, which is consistent with previous work [21,29,30,73]. Several studies have shown a lower frequency of moderate biofilm-forming strains and a higher prevalence of weak biofilm-producing strains [18,73,74]. These differences may be due to methodological differences, such as environmental conditions that may influence biofilm capacity and experimental settings [75], differences in geographical areas, study times, or the source of sample isolation [71,72,73,74,75,76].

Similar to previous work [77], strong and moderate biofilm-producing bacteria presented higher resistance to different antimicrobial agents, including ampicillin (100–83.7%), trimethoprim/sulfamethoxazole (69.2–77.8%), ampicillin-sulbactam (76.9–61.1%), cefazolin (60–53.2%), and ceftazidime (45.5–30.4%) compared to weak biofilm-producing bacteria. Tabasi et al. (2015) [62] observed ampicillin resistance rates of 77.6%, while Atray et al. (2015) [71] observed higher rates of resistance to cephalosporins (92.8–100%). In our study, although resistance to various antibiotics was generally high, biofilm-forming organisms were more MDR (69.2%) compared to weak biofilm-producing bacteria (28.6%). This is similar to the results of the study by Katongole et al., 2020 [78].

In agreement with Neupane et al., 2016 [74], and Qian et al., 2022 [79], we observed a significant increase in levels of resistance to cephalosporin antibiotics on strong biofilm-producing strains, including cefazolin, ceftazidime, and cefepime, whereas moderate biofilm-producing strains showed increased levels of resistance to ampicillin, ampicillin-sulbactam, and piperacillin-tazobactam, suggesting that the biofilm formed by these UPEC isolates provides the ability to survive when exposed to these antibiotics [3,80]. As hospital-acquired UTIs are caused by cephalosporin-resistant strains, it is crucial to regulate antibiotic overuse to limit the spread of cephalosporin-resistant microorganisms in hospital settings. Our findings suggest that strong biofilm-producing bacteria are associated with increased antibiotic resistance to beta-lactam antibiotics such as ampicillin and cephalosporins. 

The presence of virulence factors (genes) with the ability of UPEC to form biofilms in vitro has been reported [27,28,30,35,69,73]; nevertheless, the quantitative correlation between biofilm and virulence factors has yielded different results. In agreement with previous studies [21,29,30,69,74,76,80,81,82], strong biofilm-producing strains presented a higher prevalence of *fimH* and *papC* genes. Indeed, *fimH,* the gene coding for the α-D-mannose-specific tip adhesin of type 1 fimbriae, was present in almost all the strains, denoting their important role in adhesion and biofilm formation in UPEC. More than half of the strains presented *papC*, which is a colonization factor mainly expressed in pyelonephritis and has been previously correlated with strong biofilm formation [30]. In accordance with previous studies, type 1 fimbriae expression was prevalent among strong and moderate biofilm-producing strains [28,61,83], as well as curli fimbriae and cellulose. Type 1 pili is considered to be an essential virulence determinant of cystitis-causing UPEC, and it is also necessary for the intracellular aggregation of the bacteria into the IBC biofilm-like mass in the initiation and maturation process [7,43,84]. Indeed, it has been found that not-produced type 1 fimbriae by UPEC fail to form IBC in vivo, and UPEC presented attenuated virulence [84]. Curli promotes adherence of UPEC to epithelial cells derived from human bladder and kidneys [85], is an essential component of the biofilm matrix, and enhances biofilm formation by facilitating initial cell-surface and cell–cell interactions in *E. coli* [40,86]. In addition, Samet et al. [87] showed that most isolates expressing curli could produce biofilms. In our study, curli and cellulose production were more frequent among strong and moderate biofilm-producing strains isolated from hospital-acquired infections. Indeed, we found important cellulose production and curli fimbriae among strong biofilm production. Thus, these findings suggest a better ability of the strains to produce biofilm in vitro and in vivo, as well as a better capacity to produce a lower urinary tract.

There are non-significant differences between the prevalence of virulence genes tested by PCR on strong versus weak biofilm-producing bacteria. Regardless, when we compared optical densities of biofilm formation, there was a significant difference between *sfaS*-positive versus *sfaS*-negative strains and phylogroup B1 versus other phylogroups among high biofilm producers, suggesting their role in biofilm production. S fimbriae are mannose-resistant adhesins that bind to glycoproteins of urothelial tissues in the bladder and kidneys and have been commonly detected in strains that form strong-producing biofilms in vitro [27,31,40]. Additionally, the association between biofilm formation and several virulence genes has been reported as a variable since they have an enormous and variable genetic repertoire [27,83,88,89,90]. Indeed, the variation in virulence genes of *E. coli* is due to differences in the isolation of ExPEC strains in different geographical regions [76]. However, resistance to beta-lactam antimicrobials and cephalosporins is highly prevalent among strains with better biofilm-forming abilities.

In this study, the strain collection is limited to a population from a specific geographic region (Aguascalientes). Although the association between resistance to beta-lactamases and biofilm formation was established, it may not reflect the actual situation of UPEC strains within the population. Further studies involving larger clinical strains are required to determine the influence of biofilm-forming ability and antimicrobial resistance patterns on UPEC strains.

Our results suggest that a better ability to form biofilm is associated with type 1 fimbriae expression, curly, and cellulose. Other virulence genes that encode fimbriae adhesins, such as *fimH*, *papC,* and *sfaS* seem to contribute to better biofilm formation. Moreover, *E. coli* isolated from urinary tract infections that was strong biofilm-producing presented higher resistance rates to the antimicrobials ampicillin, ampicillin-sulbactam, cefazolin, ceftazidime, and cefepime.

The relationship between strong biofilm-producing strains and high resistance to beta-lactam antimicrobials and cephalosporin in the third and fourth generations and the high ability of UPEC strains to produce biofilm provided sensitive information. This should be considered during UTI treatment. Biofilm production by *E. coli* may promote colonization, increasing UTI rates. Such infections may be challenging to treat because they are associated with multidrug resistance, especially UTIs isolated from males, which have higher resistance rates to almost all antibiotics tested.

## Figures and Tables

**Figure 1 microorganisms-11-02858-f001:**
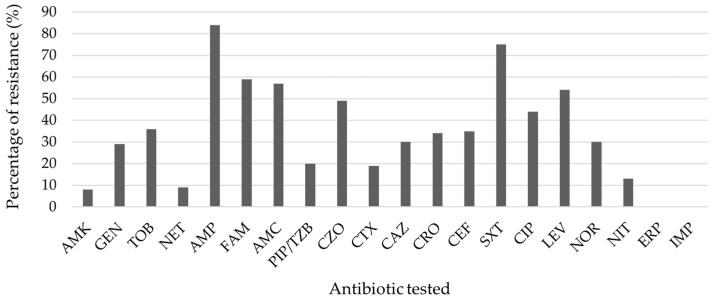
Antimicrobial resistance pattern of strains isolates from UTI patients (N = 74). AMK, amikacin; GEN, gentamicin; TOB, tobramycin; NET, netilmicin; AMP, ampicillin; FAM, ampicillin-sulbactam; AMC, amoxicillin-clavulanic acid; PIP/TZB, piperacillin-tazobactam; CZO, cefazolin; CTX, cefotaxime; CAZ, ceftriaxone; CEF, cefepime; SXT, trimethoprim-sulfamethoxazole; CIP, ciprofloxacin; LEV, levofloxacin; NOR, norfloxacin; NIT, nitrofurantoin; ERT, ertapenem; IMP, imipenem. Netilmicin (NIT) antimicrobial resistance was tested only on thirty-one samples.

**Figure 2 microorganisms-11-02858-f002:**
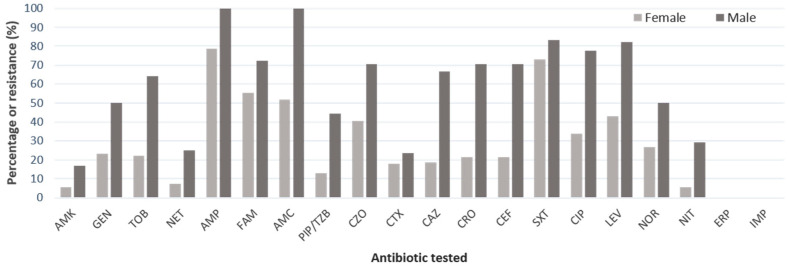
Antimicrobial resistance pattern of strains isolates from female (*n* =56) and male (*n* =18) UTI patients (N = 74). AMK, amikacin; GEN, gentamicin; TOB, tobramycin; NET, netilmicin; AMP, ampicillin; FAM, ampicillin-sulbactam; AMC, amoxicillin-clavulanic acid; PIP/TZB, piperacillin-tazobactam; CZO, cefazolin; CTX, cefotaxime; CAZ, ceftriaxone; CEF, cefepime; SXT, trimethoprim-sulfamethoxazole; CIP, ciprofloxacin; LEV, levofloxacin; NOR, norfloxacin; NIT, nitrofurantoin; ERT, ertapenem; IMP, imipenem. Netilmicin (NIT) antimicrobial resistance was tested only on thirty-one samples.

**Figure 3 microorganisms-11-02858-f003:**
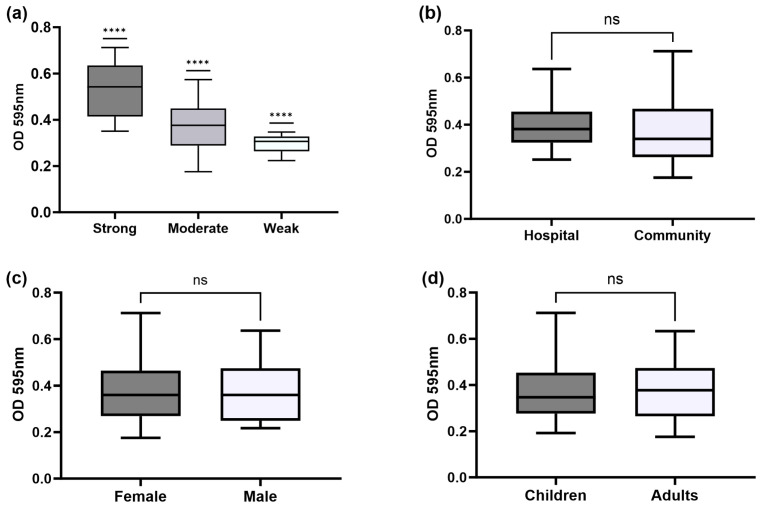
Distribution of biofilm formation among characteristics of the patients. Biofilm formation was determined by crystal violet assay for *E. coli* isolated from urinary tract infections (UTIs). (**a**) Optical density (595 nm) among strong, moderate, and weak biofilm formation; (**b**) biofilm formation among hospital (*n* = 23) and community-acquired infection (*n* = 51); (**c**) female (*n* = 56) and males (*n* = 18); and (**d**) children (*n* = 45) and adults (*n* = 29). A higher optical density (OD) for each box plot indicates higher adhesion and biofilm formation. The data are presented as box plots, and the whiskers extend to the minimum and maximum values. A higher optical density (OD) for each box plot indicates higher adhesion and biofilm formation. Only significant *p*-values are shown (**** *p <* 0.0001), ns: non-statistical significance. Mann–Witney U test, two-tailed, was used to compare optical density mean (OD) among characteristics of the patients.

**Figure 4 microorganisms-11-02858-f004:**
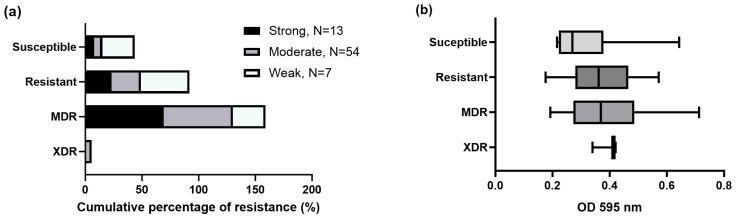
Resistance levels of different antibiotics tested among strong, moderate, and weak biofilm-producing strains. (**a**) Cumulative percentage of resistance, and (**b**) resistance levels among optical density (OD 595 nm). The data are presented as box plots, and the whiskers extend to the minimum and maximum values. A higher optical density (OD) for each box plot indicates higher adhesion and biofilm formation.

**Figure 5 microorganisms-11-02858-f005:**
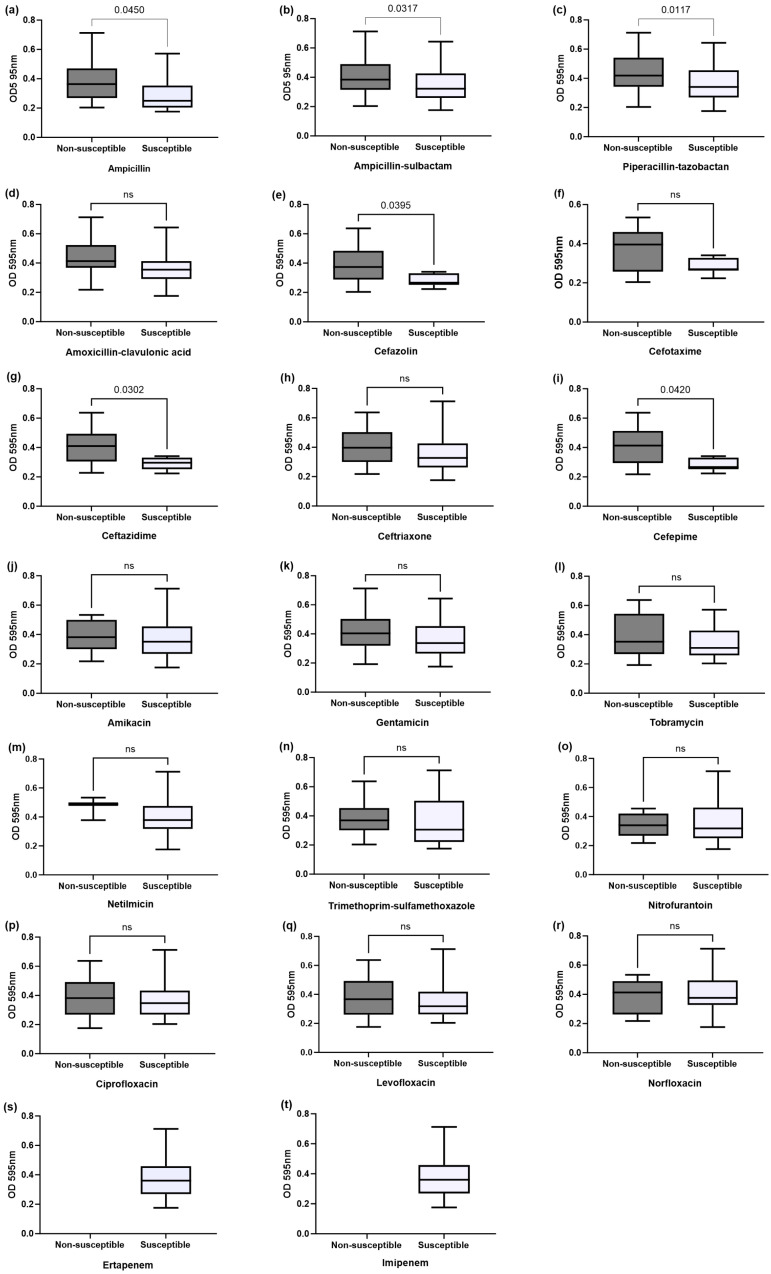
Distribution of biofilm formation among the different antibiotic resistance profiles. (**a**) Ampicillin; (**b**) ampicillin-sulbactam; (**c**) piperacillin-tazobactam; (**d**) amoxicillin-clavulanic acid; (**e**) cefazolin; (**f**) cefotaxime; (**g**) ceftazidime; (**h**) ceftriaxone; (**i**) cefepime; (**j**) amikacin; (**k**) gentamicin; (**l**) tobramycin; (**m**) netilmicin; (**n**) trimethoprim-sulfamethoxazole; (**o**) nitrofurantoin; (**p**) ciprofloxacin; (**q**) levofloxacin; (**r**) norfloxacin; (**s**) ertapenem; (**t**) imipenem. The data are presented as box plots, and the whiskers extend to the minimum and maximum values. A higher optical density (OD) for each box plot indicates higher adhesion and biofilm formation. Netilmicin susceptibility was tested only on thirty-one samples. Mann–Witney U test, two-tailed, was used to compare optical density mean (OD) among non-susceptible vs. susceptible strains to different antimicrobial agents.

**Table 1 microorganisms-11-02858-t001:** Comparison among virulence genes, phylogenetic groups, and antimicrobial resistance patterns of *Escherichia coli* isolates from urinary tract infections.

Category		Total *n* = 74 (%)	Hospital-Acquired *n* = 23 (%)	Community-Acquired *n* = 51 (%)	** p*-Value
Gender	Female	56 (75.7)	15 (65.2)	41 (80.4)	
	Male	18 (24.3)	8 (34.8)	10 (19.6)	
Age	Children	45 (60.8)	10 (43.5)	35 (68.3)	0.0402
	Adults	29 (39.2)	13 (56.5)	16 (31.4)	
Antimicrobial resistance	Ceftazidime resistance	19 (25.7)	10 (43.5)	9 (17.6)	0.0355
	Susceptible	7 (9.5)	2 (8.7)	5 (9.8)	
	Resistant	67 (90.5)	21 (91.3)	46 (90.2)	
	Multidrug-resistant	47 (63.5)	14 (60.9)	33 (64.7)	
Virulence gene	*fimH*	65 (87.8)	20 (87.9)	45 (88.2)	
	*papC*	40 (54.1)	14 (16.9)	26 (51.0)	
	*sfaS*	10 (13.5)	2 (8.7)	8 (15.7)	
	*afa/Dr*	9 (12.2)	2 (8.7)	7 (13.7)	
	*yfcV*	21 (28.4)	8 (34.8)	13 (24.5)	
	*agn43*	53 (71.6)	15 (65.2)	38 (74.5)	0.0220
	*vat*	21 (28.4)	5 (21.7)	16 (31.4)	
	*cnf1*	3 (4.1)	1 (4.3)	2 (3.9)	
	*hlyA*	9 (12.2)	3 (13.0)	6 (11.8)	
	*fyuA*	59 (79.7)	19 (82.6)	40 (78.4)	
	*chuA*	49 (66.2)	14 (60.9)	35 (68.6)	
	*kpsMTII*	38 (51.4)	16 (69.6)	22 (43.1)	0.0341
Virulence score		5.09 ± 1.58	5.17 ± 1.66	5.06 ± 1.55	
Phylogenetic group	A	4 (5.4)	1 (4.5)	3 (5.9)	
	B1	9 (12.2)	5 (21.7)	4 (7.8)	
	B2	12 (16.2)	3 (13.0)	9 (17.6)	
	C	9 (12.2)	3 (13.0)	6 (11.8)	
	D	22 (29.7)	4 (17.4)	18 (35.3)	
	E	1 (1.4)	1 (4.3)	0 (0.0)	
	F	11 (14.9)	4 (17.4)	7 (13.7)	
	Clades	6 (8.1)	2 (8.7)	4 (7.8)	

* Only significant *p* values are shown (*p* < 0.05). Categorical variables were tested by chi-squared and Fisher exact test. Mann–Witney U test, two-tailed, was used to compare continuous variables among hospital- and community-acquired infections.

**Table 2 microorganisms-11-02858-t002:** Comparison among virulence genes, phylogenetic groups, and antimicrobial resistance patterns of *Escherichia coli* isolates from female and male urinary tract infections.

Category		Total *n* = 74 (%)	Female Patients *n* = 56 (%)	Male Patients *n* = 18 (%)	** p*-Value
Age	Children	45 (60.8)	37 (66.07)	8 (44.44)	
	Adults	29 (39.2)	8 (14.28)	10 (55.55)	
Antimicrobial resistance	XDR and MDR	47 (63.5)	30 (53.6)	17 (94.4)	0.0017
	XDR	3 (5.1)	1 (1.8)	2 (11.1)	
	MDR	44 (59.5)	29 (51.8)	15 (83.3)	0.0261
	R	20 (27)	19 (33.9)	1 (56.6)	0.0297
	Susceptible	7 (9.5)	7 (12.5)	0 (0.0)	
Virulence gene	*fimH*	65 (87.8)	47 (83.9)	18 (100)	
	*papC*	40 (54.1)	29 (51.8)	11 (61.1)	
	*sfaS*	10 (13.5)	8 (14.3)	2 (11.1)	
	*afa/Dr*	9 (12.2)	9 (16.1)	0 (0.0)	
	*yfcV*	21 (28.4)	12 (21.4)	9 (50.0)	0.0193
	*agn43*	53 (71.6)	40 (71.4)	13 (72.2)	
	*vat*	21 (28.4)	14 (25)	7 (38.9)	
	*cnf1*	3 (4.1)	2 (3.6)	1 (5.6)	
	*hlyA*	9 (12.2)	6 (10.7)	3 (16.7)	
	*fyuA*	59 (79.7)	47 (83.9)	12 (66.7)	
	*chuA*	49 (66.2)	39 (69.6)	10 (55.6)	
	*kpsMTII*	38 (51.4)	31 (55.4)	7 (38.9)	
Virulence score		5.09 ± 1.58	5.07 ± 1.52	5.17 ± 1.79	
Phylogenetic group	A	4 (5.4)	2 (3.6)	2 (11.1)	
	B1	9 (12.2)	6 (10.7)	3 (16.7)	
	B2	12 (16.2)	7 (12.5)	5 (27.8)	
	C	9 (12.2)	6 (10.7)	3 (16.7)	
	D	22 (29.7)	21 (37.5)	1 (5.6)	0.0087
	E	1 (1.4)	1 (1.8)	0 (0.0)	
	F	11 (14.9)	9 (16.1)	2 (11.1)	
	Clades	6 (8.1)	4 (7.1)	2 (11.1)	

* Only significant *p* values are shown (*p* < 0.05). Categorical variables were tested by chi-squared and Fisher exact test. Mann–Witney U test, two-tailed, was used to compare continuous variables among female and male infections.

**Table 3 microorganisms-11-02858-t003:** Phylogroups and virulence genes profile distribution among strong, moderate, and weak biofilm-producing *E. coli* isolates from urinary tract infections.

Virulence Genes and Phylogroups		Biofilm-Producing Abilities			
		Total *n* = 74 (%)	Strong *n* = 13 (%)	Moderate *n* = 54 (%)	Weak *n* = 7 (%)
Virulence gene	*fimH*	65 (87.8)	11 (84.6)	47 (87.0)	7 (100)
	*papC*	40 (54.1)	10 (76.9)	26 (48.1)	4 (57.1)
	*sfaS*	10 (13.5)	2 (15.4)	7 (12.9)	1 (14.3)
	*afa/Dr*	9 (12.2)	2 (15.4)	6 (11.1)	1 (14.3)
	*yfcV*	21 (28.4)	5 (38.5)	14 (25.9)	2 (28.6)
	*agn43*	53 (71.6)	9 (69.2)	39 (72.2)	5 (71.4)
	*vat*	21 (28.4)	3 (23.1)	16 (29.6)	2 (28.6)
	*cnf1*	3 (4.1)	1 (7.7)	1 (1.9)	1 (14.3)
	*hlyA*	9 (12.2)	2 (15.4)	7 (12.9)	0 (0.0)
	*fyuA*	59 (79.7)	9 (69.2)	44 (81.5)	6 (85.7)
	*chuA*	49 (66.2)	9 (69.2)	34 (62.9)	6 (85.7)
	*kpsMTII*	38 (51.4)	5 (35.5)	29 (53.7)	4 (57.1)
Virulence score		5.09 ± 1.58	5.23 ± 1.79	5.00 ± 1.57	5.57 ± 1.40
Phylogenetic group	A	4 (5.4)	1 (7.7)	3 (5.6)	0 (0.0)
	B1	9 (12.2)	1 (7.7)	8 (14.8)	0 (0.0)
	B2	12 (16.2)	1 (7.7)	9 (16.7)	2 (28.6)
	C	9 (12.2)	1 (7.7)	7 (12.9)	1 (14.3)
	D	22 (1.4)	4 (30.8)	14 (25.7)	4 (57.1)
	E	1 (1.4)	0 (0.0)	1 (1.9)	0 (0.0)
	F	11 (14.9)	3 (23.1)	8 (14.8)	0 (0.0)
	Clades	6 (8.2)	2 (15.4)	4 (7.4)	0 (0.0)

Non-significant *p* values were found among categories. Comparison among strong, moderate, and weak biofilms was performed by a non-parametrical one-way ANOVA with Dunn´s multiple comparisons test.

**Table 4 microorganisms-11-02858-t004:** Phenotypically expressed surface virulence factors and multidrug-resistant patterns among strong, moderate, and weak biofilm-producing *E. coli* isolates from urinary tract infections.

Phenotypically Expressed Virulence Factors	Biofilm-Producing Abilities				
	Total *n* = 74 (%)	Strong *n* =13 (%)	Moderate *n* = 54 (%)	Weak *n* = 7 (%)	* *p*-Value
Type 1 fimbriae (MSHA) ^a^	55 (74.3)	11 (84.6)	40 (74.1)	4 (57.1)	0.0349
P fimbriae (MRHA) ^b^	19 (25.7)	2 (15.4)	14 (25.9)	3 (42.9)	
α-Hemolysis	53 (71.6)	9 (69.2)	39 (72.2)	5 (71.4)	
Curli fimbriae	48 (64.9)	8 (61.5)	37 (68.5)	3 (42.9)	0.0477
Cellulose	41 (55.4)	9 (69.2)	30 (55.5)	2 (25.7)	0.0253

* Only significant *p* values are shown (*p* < 0.05). ^a^ MSHA—mannose-sensitive hemagglutination; ^b^ MRHA—mannose-resistant hemagglutination. Curli production as estimated by calcofluor binding.

## Data Availability

Data are contained within the article.

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
