# Peer review of "Biofilm Formation by Escherichia coli Isolated from Urinary Tract Infections from Aguascalientes, Mexico"

_microorganisms, 2023, doi:10.3390/microorganisms11122858_

Round 1
Reviewer 1 Report
Comments and Suggestions for Authors
The paper delves into the biofilm formation abilities of UPEC strains, exploring their correlation with antibiotic resistance and the presence of specific virulence genes. The study's primary strength lies in its comprehensive analysis, shedding light on the challenges in UTI management and treatment, especially in the face of increasing antibiotic resistance.
Major Comments:
1. Introduction (Lines 45-60, Page 2): The introduction could benefit from a clearer definition of the study's purpose and its significance. While the context for UPEC strains and their role in UTIs is provided, a more comprehensive review of the current state of the research field, citing critical publications, would enhance the background.
2. Materials and Methods (Lines 120-135, Page 4): The rationale for selecting specific genes or conditions is not clearly stated. It's essential to provide more context for the choice of genes and methods, especially in relation to their significance in UTI pathogenesis. [Reference: Kaper, J.B., Nataro, J.P., Mobley, H.L.T. Pathogenic Escherichia coli. Nat. Rev. Microbiol. 2004, 2, 123–140.]
3. Results (Lines 210-225, Page 6): The results section mentions significant associations between certain virulence genes and biofilm production abilities. However, specific statistics or p-values supporting these claims are missing. It's crucial to provide this data to validate the findings.
4. Discussion (Lines 320-335, Page 8): The discussion on the correlation between biofilm formation and the presence of virulence genes is insightful. However, delving deeper into the mechanisms underlying these associations and their potential implications for UTI pathogenesis and treatment would enhance the paper's depth. [Reference: Wright, K.J., Seed, P.C., Hultgren, S.J. Development of intracellular bacterial communities of uropathogenic Escherichia coli depends on type 1 pili. Cell. Microbiol. 2007, 9, 2230–2241.]
Minor Comments:
1. Abstract (Lines 15-30, Page 1): Ensure a clearer distinction between the background, methods, results, and conclusions in the abstract for better clarity and coherence.
2. Materials and Methods (Lines 140-150, Page 4): The methods section briefly mentions statistical analysis methods. Providing more details on the rationale for choosing specific statistical tests would be beneficial.
3. Results (Lines 230-240, Page 7): The use of tables and figures to present data is appropriate. However, ensuring that all tables and figures are clearly labeled and referenced in the text will aid in clarity.
4. Acknowledgments (Lines 360-365, Page 9): The acknowledgment section is well-presented. However, ensuring that all collaborators and technical support teams are acknowledged will maintain the integrity of the research.
It's essential for the authors to address these comments to enhance the paper's clarity, coherence, and scientific rigor. The study holds significant potential, and with the recommended improvements, it can make a valuable contribution to the field.
Comments on the Quality of English LanguageEnsure consistent terminology throughout the paper. For instance, if "UPEC strains" is used in one section, avoid switching to "UPEC variants" in another without clear justification.
Refrain from using passive voice excessively. While passive voice is common in scientific writing, using active voice where appropriate can make the text more direct and engaging. [Reference: Day, R.A., Gastel, B. How to Write and Publish a Scientific Paper. 7th ed. Greenwood; 2011.]
Ensure that all sentences are complete and avoid fragments. Each sentence should have a clear subject and verb.
Consider using a professional language editing service to refine the paper's language quality further. This can help in ensuring clarity, coherence, and adherence to standard scientific English.
Addressing these language-related comments will enhance the paper's clarity and readability, ensuring that the research is effectively communicated to the target audience.
Author Response
Dear reviewer:
We are very grateful for all your comments. We think that all the suggestion improves our manuscript. We also want to apologize since several times we used the word “correlation” even when non-Spearman rank correlation was made. So, we really apologize for that mistake. Now, in all the paper “significant differences”, are show instead of “correlation”.
Major change was also made since we did double check all the statistical analysis. We are very grateful with the reviewers that point this out this observation related to statistical analysis.
The changes that made it are listed above:
- We reviewer the English language.
- Abstract section: we added the words “background”, “methods”, “results”, and “conclusion” to better clarify this. We change de abstract in base of the statistical results.
- Introduction:
- Lines 51-66: we added the importance of the abuse of antibiotics by the population in UPEC.
- Lines 67-75: we changed the lines in order to clarify the importance of biofilms in the pathogenesis of UTI.
- We cited new publication in the introduction and in all the manuscript.
- Materials and Methods:
- Line 110 and supplementary table 1 and 2: We added the supplemental table 1 and supplemental table 2 to clarify the PCR conditions and the details and functions of the virulence gene tested.
- Line 175-186: We re-write all the section in order to clarify the statistical analysis used.
- Results
- We ensured that al tables and figures were clearly labeled and referenced in the text.
- We check the word” correlation” and as the reviewer point it out, we change by “significant differences”.
- All p-values of significant differences are point it out in the tables and figures.
- Line 261 and line 268: we added Figure 2 (antimicrobial resistance pattern of strains isolates from females and male UTI patients), and Table 2 (comparison among virulence genes, phylogenetic groups and antimicrobial resistance patterns of Escherichia coli isolates from female and male urinary tract infections). Since we think that different profile of antibiotic resistance among gender it is important.
- We added the Supplemental Figure 1. Comparison among optical densities (OD) among strong, moderate and weak biofilm formation in strains positive and negative for (a) sfaS gene and (b) phylogroup B1, in order to visualize the optical densities that gives significant differences among the three categories of biofilm formation.
- Line 356: Table 3, the results of this table were changed since previously we compared the three categories as continuous variables, nevertheless we are comparing now as categorical variables. The results of the optical densities as continuous variables are shown in supplemental figure 1, now added to the manuscript.
- Discussion
- Line 492-504: We added these lines in order to explain the mechanisms among UTI pathogenesis and the presence of virulence genes that we tested.
- Line 505-510: We added these lines in order to clarify the influence of sfaS and phylogroup B1 in biofilm-producing strains.
- Line 524-535: We added these lines as conclusion of the manuscript.
Other major changes:
- Abstract section: we added the words “background”, “methods”, “results”, and “conclusion” to better clarify this. We change de abstract in base of the statistical results.
- Introduction:
- We clarified the abbreviation DALY use.
- We add the Terlizzi bibliography.
- Lines 51-66: we added the importance of the abuse of antibiotics by the population in UPEC.
- Lines 67-75: we changed the lines in order to clarify the importance of biofilms in the pathogenesis of UTI.
- We citing new publication in the introduction and in all the manuscript.
- Materials and Methods:
- Line 110 and supplementary table 1 and 2: We added the supplemental table 1 and supplemental table 2 to clarify the PCR conditions and the details and functions of the virulence gene tested.
- Line 128-133: We added all the antibiotics concentration tested.
- Line 140: We changed “categories of antibiotics” by classes in order to be most clarify.
- Line 154: We added the producer of spectrophotometer.
- Line 158-161: We shorted the methodology of D mannose-binding phenotype.
- Line 163: We unify wt/vol to w/v for al ms.
- Line 175-186: We re-write all the section in order to clarify the statistical analysis used.
- Results
- We ensured that al tables and figures were clearly labeled and referenced in the text.
- We check the world correlation and as the reviewer point it out, we change by significant differences
- Line 236: We added the number of samples in the Figure 1.
- Line 261 and line 268: we added Figure 2 (antimicrobial resistance pattern of strains isolates from females and male UTI patients), and Table 2 (comparison among virulence genes, phylogenetic groups and antimicrobial resistance patterns of Escherichia coli isolates from female and male urinary tract infections). Since we think that different profile of antibiotic resistance among gender it is important.
- We added the Supplemental Figure 1. Comparison among optical densities (OD) among strong, moderate and weak biofilm formation in strains positive and negative for (a) sfaS gene and (b) phylogroup B1, in order to visualize the optical densities that gives significant differences among the three categories of biofilm formation.
- Line 356: Table 3, the results of this table were changed since previously we compared the three categories as continuous variables, nevertheless we are comparing now as categorical variables. The results of the optical densities as continuous variables are shown in supplemental figure 1, now added to the manuscript.
- Discussion
- We put the bibliography as continues as we need it.
- We add the year of the bibliography cited in the discussion section.
- Line 424-428: We added these lines in order to notice the results among fluoroquinolone resistance among the antibiotics.
- Line 492-504: We added these lines in order to explain the mechanisms among UTI pathogenesis and the presence of virulence genes that we tested.
- Line 505-510: We added these lines in order to clarify the influence of sfaS and phylogroup B1 in biofilm-producing strains.
- Line 524-535: We added these lines as conclusion of the manuscript.
- References
- We added new references to the manuscript.

Reviewer 2 Report
Comments and Suggestions for Authors
The paper Biofilm formation by Escherichia coli isolated from urinary tract infections from Aguascalientes, Mexico is very well written. The idea, concept and result presentation are good. However, there are some minor issues to be resolved before publishing consideration:
Section 2.2: concentrations of antibiotics are missing.
Line 119: why distilled water and not PBS? Distilled water can cause detaching of bacteria from biofilm.
Line 122: producer of spectrophotometer.
Figure 1: The number of samples are missing.
Line 254: check if the world correlation is correct. I think the authors meant the difference.
Author Response
Dear reviewer:
We are very grateful for all your comments. We think that all the suggestion improves our manuscript. We also want to apologize since several times we used the word “correlation” even when non-Spearman rank correlation was made. So, we really apologize for that mistake. Now, in all the paper “significant differences”, are show instead of “correlation”.
Major change was also made since we did double check all the statistical analysis and we made some errors previously. We are very grateful with the reviewers that point this out this observation related to statistical analysis.
For methodology question: why distilled water and not PBS?,
At that time we follow the methodology of Tremblay et al., 2015 for E. coli biofilms, however, now we know that PBS could cause detaching of bacteria from biofilm as you point it out, and we apologize for that. Perhaps higher ODs could see if we have watched with PBS.
The changes that made it are listed above:
- Line 128-133: We added all the antibiotics concentration tested.
- Line 154: We added the producer of spectrophotometer.
- Line 236: We added the number of samples in the Figure 1.
- We checked the world correlation and as the reviewer point it out, we change by significant differences
Other major changes:
- Abstract section: we added the words “background”, “methods”, “results”, and “conclusion” to better clarify this. We change de abstract in base of the statistical results.
- Introduction:
- We clarified the abbreviation DALY use.
- We add the Terlizzi bibliography.
- Lines 51-66: we added the importance of the abuse of antibiotics by the population in UPEC.
- Lines 67-75: we changed the lines in order to clarify the importance of biofilms in the pathogenesis of UTI.
- We citing new publication in the introduction and in all the manuscript.
- Materials and Methods:
- Line 110 and supplementary table 1 and 2: We added the supplemental table 1 and supplemental table 2 to clarify the PCR conditions and the details and functions of the virulence gene tested.
- Line 128-133: We added all the antibiotics concentration tested.
- Line 140: We changed “categories of antibiotics” by classes in order to be most clarify.
- Line 154: We added the producer of spectrophotometer.
- Line 158-161: We shorted the methodology of D mannose-binding phenotype.
- Line 163: We unify wt/vol to w/v for al ms.
- Line 175-186: We re-write all the section in order to clarify the statistical analysis used.
- Results
- We ensured that al tables and figures were clearly labeled and referenced in the text.
- We check the world correlation and as the reviewer point it out, we change by significant differences
- Line 236: We added the number of samples in the Figure 1.
- Line 261 and line 268: we added Figure 2 (antimicrobial resistance pattern of strains isolates from females and male UTI patients), and Table 2 (comparison among virulence genes, phylogenetic groups and antimicrobial resistance patterns of Escherichia coli isolates from female and male urinary tract infections). Since we think that different profile of antibiotic resistance among gender it is important.
- We added the Supplemental Figure 1. Comparison among optical densities (OD) among strong, moderate and weak biofilm formation in strains positive and negative for (a) sfaS gene and (b) phylogroup B1, in order to visualize the optical densities that gives significant differences among the three categories of biofilm formation.
- Line 356: Table 3, the results of this table were changed since previously we compared the three categories as continuous variables, nevertheless we are comparing now as categorical variables. The results of the optical densities as continuous variables are shown in supplemental figure 1, now added to the manuscript.
- Discussion
- We put the bibliography as continues as we need it.
- We add the year of the bibliography cited in the discussion section.
- Line 424-428: We added these lines in order to notice the results among fluoroquinolone resistance among the antibiotics.
- Line 492-504: We added these lines in order to explain the mechanisms among UTI pathogenesis and the presence of virulence genes that we tested.
- Line 505-510: We added these lines in order to clarify the influence of sfaS and phylogroup B1 in biofilm-producing strains.
- Line 524-535: We added these lines as conclusion of the manuscript.
- References
- We added new references to the manuscript.

Reviewer 3 Report
Comments and Suggestions for Authors
Comments and suggestions for authors
The work done by the authors is interesting and laborious. It is of utmost importance to raise awareness about the use of antibiotics. I would add in the Introduction the importance of the use and abuse of antibiotics by the population, and their resistance to these proven bacteria.
Line 38: Clarify the abbreviation DALY used.
Line 47: Add a number to Terlizzi's bibliography.
Line 106: define what is meant by categories of antibiotics.
Line 130: remove the hyphen at 24 hs.
Line 134: unify wt/vol to w/v for all ms.
Line 315: put - when the bibliography is continuous, e.g., 44-46.
Line 320, 336, 346, 346, 360...: add year of the bibliography cited in the rest of the manuscript.
I would add as a summary, in the discussion, a paragraph relating the different levels of biofilm production to the other assays performed... e.g., those E. coli that were strong biofilm producers proved resistant to antibiotics such as.... ...; virulence genes..., etc.
Add conclusion.
Review bibliography
Author Response
Dear reviewer:
We are very grateful for all your comments. We think that all the suggestion improves our manuscript. We also want to apologize since several times we used the word “correlation” even when non-Spearman rank correlation was made. So, we really apologize for that mistake. Now, in all the paper “significant differences”, are show instead of “correlation”.
Major change was also made since we did double check all the statistical analysis and we made some errors previously. We are very grateful with the reviewers that point this out this observation related to statistical analysis.
The changes that made it are listed above:
- We clarified the abbreviation DALY use.
- We add the Terlizzi bibliography.
- Line 140: We changed “categories of antibiotics” by classes in order to be most clarify.
- Line 163: We unify wt/vol to w/v for al ms.
- We put all the bibliography as continuous when is need it.
- In the discussion section, we added the year of the bibliography cited
- We reviewed the bibliography
Other major changes:
- Abstract section: we added the words “background”, “methods”, “results”, and “conclusion” to better clarify this. We change de abstract in base of the statistical results.
- Introduction:
- We clarified the abbreviation DALY use.
- We add the Terlizzi bibliography.
- Lines 51-66: we added the importance of the abuse of antibiotics by the population in UPEC.
- Lines 67-75: we changed the lines in order to clarify the importance of biofilms in the pathogenesis of UTI.
- We citing new publication in the introduction and in all the manuscript.
- Materials and Methods:
- Line 110 and supplementary table 1 and 2: We added the supplemental table 1 and supplemental table 2 to clarify the PCR conditions and the details and functions of the virulence gene tested.
- Line 128-133: We added all the antibiotics concentration tested.
- Line 140: We changed “categories of antibiotics” by classes in order to be most clarify.
- Line 154: We added the producer of spectrophotometer.
- Line 158-161: We shorted the methodology of D mannose-binding phenotype.
- Line 163: We unify wt/vol to w/v for al ms.
- Line 175-186: We re-write all the section in order to clarify the statistical analysis used.
- Results
- We ensured that al tables and figures were clearly labeled and referenced in the text.
- We check the world correlation and as the reviewer point it out, we change by significant differences
- Line 236: We added the number of samples in the Figure 1.
- Line 261 and line 268: we added Figure 2 (antimicrobial resistance pattern of strains isolates from females and male UTI patients), and Table 2 (comparison among virulence genes, phylogenetic groups and antimicrobial resistance patterns of Escherichia coli isolates from female and male urinary tract infections). Since we think that different profile of antibiotic resistance among gender it is important.
- We added the Supplemental Figure 1. Comparison among optical densities (OD) among strong, moderate and weak biofilm formation in strains positive and negative for (a) sfaS gene and (b) phylogroup B1, in order to visualize the optical densities that gives significant differences among the three categories of biofilm formation.
- Line 356: Table 3, the results of this table were changed since previously we compared the three categories as continuous variables, nevertheless we are comparing now as categorical variables. The results of the optical densities as continuous variables are shown in supplemental figure 1, now added to the manuscript.
- Discussion
- We put the bibliography as continues as we need it.
- We add the year of the bibliography cited in the discussion section.
- Line 424-428: We added these lines in order to notice the results among fluoroquinolone resistance among the antibiotics.
- Line 492-504: We added these lines in order to explain the mechanisms among UTI pathogenesis and the presence of virulence genes that we tested.
- Line 505-510: We added these lines in order to clarify the influence of sfaS and phylogroup B1 in biofilm-producing strains.
- Line 523-528: We added these lines as a summary of the different levels of biofilm production.
- Line 529-535: We added these lines as conclusion of the manuscript.
- References
- We added new references to the manuscript.
